# Graduated Assignment for Joint Multi-Graph Matching and Clustering with Application to Unsupervised Graph Matching Network Learning

**Runzhong Wang**[1,2]     **Junchi Yan**[1,2] *     **Xiaokang Yang**[2]

[1] Department of Computer Science and Engineering, Shanghai Jiao Tong University
[2] MoE Key Lab of Artificial Intelligence, AI Institute, Shanghai Jiao Tong University
`{runzhong.wang,yanjunchi,xkyang}@sjtu.edu.cn`

## Abstract

This paper considers the setting of jointly matching and clustering multiple graphs belonging to different groups, which naturally rises in many realistic problems. Both graph matching and clustering are challenging (NP-hard) and a joint solution is appealing due to the natural connection of the two tasks. In this paper, we resort to a graduated assignment procedure for soft matching and clustering over iterations, whereby the two-way constraint and clustering confidence are modulated by two separate annealing parameters, respectively. Our technique can be further utilized for end-to-end learning whose loss refers to the cross-entropy between two lines of matching pipelines, as such the keypoint feature extraction CNNs can be learned without ground-truth supervision. Experimental results on real-world benchmarks show our method outperforms learning-free algorithms and performs comparatively against two-graph based supervised graph matching approaches. Source code is publicly available as a module of `ThinkMatch`.

## 1   Introduction and Related Works

Graph matching (GM) computes the node-to-node matching relationship among multiple graphs, by utilizing the structural information in graphs, which can be applied to align various real-world graphs. GM is in general NP-hard [1], and can be formulated as Quadratic Assignment Problem (QAP).

Traditional GM algorithms mainly solve two-graph matching problems via continuous relaxation [2, 3, 4, 5, 6] or tree search algorithms [7, 8] with fixed affinity metric, while there exists increasing interest on developing more robust graph matching by fusing multiple graph information, known as multi-graph matching (MGM) [9, 10, 11, 12, 13, 14] and adopting learnable affinity functions (known as learning graph matching [15, 16], especially deep graph matching [17, 18, 19]). However, learning, especially deep learning is merely considered in existing multi-graph matching algorithms. In addition, current supervised deep graph matching requires costly annotation on large-scaled training data, which restricts the real-world application of modern deep graph matching methods. Motivated by the fact that MGM will usually result in better matching results compared to pairwise matching [10], it is appealing to devise an unsupervised deep multi-graph matching learning algorithm by utilizing multi-graph matching information as the pseudo label for pairwise matchings.

A realistic yet more challenging setting is considered namely multi-graph matching and clustering (MGMC), where graphs may belong to different groups and clustering and matching both need to be solved. This problem has seldom been considered apart from one loosely relevant work [20], which is learning-free, and clustering is performed after matching. In contrast, observing the fact that

matching and clustering are inherently interleaved to each other, an MGMC approach is developed where clustering and matching are performed alternatively with gradually increased confidence. Furthermore, the unsupervised learning approach is still applicable to MGMC, showing the potential of real-world applications of our method. **This paper has made the following contributions:**

1) A graduated assignment multi-graph matching (GA-MGM) solver is proposed by iteratively solving the first-order Taylor expansion of MGM Koopmans-Beckmann's QAP. It is achieved by Sinkhorn annealed by parameter $\tau$ and followed by Hungarian method for the final discrete solution.

2) We extend our technique to the challenging multi-graph matching and clustering (MGMC) setting that matching graphs from different categories. Our GA-MGMC incorporates soft matching and clustering with graduated assignment, by additionally involving annealing parameter $\beta$ representing the confidence of clustering. This is in contrast to the recent effort [20] that devises a clustering-aware MGM method, whereby clustering is performed independently after matching.

3) We devise the first (to our best knowledge) unsupervised deep graph matching network, namely graduated assignment neural network (GANN), whose supervision can be either from MGM or MGMC. In GANN, the CNNs accounting for image keypoint feature extraction are trained by the cross-entropy loss between the network's prediction and the pseudo ground truth from GA-MGM/C.

4) Experimental results on real-world datasets show the efficacy and robustness of our method. Its performance can even be on par with those by supervised deep learning for two graph matching.

**Related Works.** Closely related works on graduated assignment, multi-graph matching, and learning graph matching are reviewed here. Readers are referred to [21] for a comprehensive review.

**1) Graduated Assignment.** The classical graduated assignment (GA) is originally proposed to approximately solve higher-order assignment, with applications to challenging combinatorial problems, e.g. two graph matching [6], multi-graph matching [22, 14] and traveling salesman problem [23]. GA works with the partial derivative of the original objective function, which can be computed via Taylor expansion, and the resulting linear assignment problem is solved approximately with regularized Sinkhorn algorithm [24, 25]. In this paper, the recent advances in neural network motivate us to develop a graduated assignment neural network (GANN) for two challenging real-world combinatorial tasks – MGM and MGMC.

**2) Multi-Graph Matching and Clustering.** MGM arises in scenarios where more than two graphs should be matched jointly, e.g. analyzing video sequences, some authors propose to optimize matching based on a joint matching objective function [26, 9, 27], and others aim to develop a post-processing technique to recover global consistency based on existing pairwise matching results [11, 28, 29, 30]. More realistic settings are also considered in recent literature, e.g. online incremental matching [31, 27], joint matching and partition [32], joint link prediction and matching [33]. The MGMC problem proposed by [20] can be regarded as a realistic extension from MGM, where graphs may belong to different categories. Both MGM and MGMC are considered in this paper.

Specifically, [14] proposes a graduated assignment MGM algorithm, yet a 'prototype' graph is required as the anchor (in their paper they use the first graph as the 'prototype'). This means they assume the bijection between each pair of graphs which is hard to satisfy in practice. In contrast, our proposed method is fully decentralized free from such a constraint, and it can therefore handle the setting of partial matching, i.e. only part of the nodes in each graph can find their correspondence from the other graphs. This setting is more practical and challenging. [34] develops a novel MGM algorithm, but it requires precise initialization by existing MGM methods (e.g. [29]) and does not work with a strict QAP formulation. In contrast, our method is initialization-free and tackles Koopmans-Beckmann's QAP with theoretical groundings. For the MGMC task, [20] performs clustering after matching, while our algorithm solves clustering and matching simultaneously. Finally, learning is rarely considered in existing MGM algorithms, not to mention the more challenging MGMC problem, while our method handles both MGM and MGMC with unsupervised learning.

**3) Graph Matching Neural Networks.** Inspired by recent advances in deep learning, tackling graph matching with deep networks is receiving increasing attention. The first line of work adopts deep feature extractors, e.g. VGG16 [35], with which graph matching problem is solved with differentiable solvers, resulting in an end-to-end learning scheme, where convolutional neural nets [17], graph neural nets [18, 36, 37] and geometric learning [19, 37] are incorporated. Based on the recent efforts in learning combinatorial solvers [38, 39], another line of work aims to solve the combinatorial graph

matching problem via deep learning, for both Koopmans-Beckmann's QAP [40, 18] and Lawler's QAP [41]. Our method falls within the efforts in learning a deep feature extractor, e.g. a deep CNN for images. One major issue in existing deep graph matching methods is that they are all based on supervised learning, therefore expensive annotation is required for training. In this paper, an unsupervised learning approach is developed for deep graph matching.

## 2 The Proposed Approach

In this section, we firstly introduce our proposed GA-MGM solver in Sec. 2.1. Its extension to MGMC problem is presented in Sec. 2.2. Finally, Sec. 2.3 gives the unsupervised learning scheme.

### 2.1 Graduated Assignment Multi-Graph Matching

#### 2.1.1 The Multi-Graph Matching Problem

In this paper, we consider $\mathcal{G}_1, \mathcal{G}_2, \ldots, \mathcal{G}_m$. Each graph corresponds to an instance in image and all graphs belong to the same category. $\mathcal{G}_i$ contains $n_i$ keypoints with known positions in its corresponding image. For a given graph $\mathcal{G}_i = (\mathbf{F}_i, \mathbf{A}_i)$, $\mathbf{F}_i \in \mathbb{R}^{n_i \times l}$ represents $l$-dimensional feature of $n_i$ nodes which is extracted by deep CNN and bi-linearly interpolated from images, adjacency matrix $\mathbf{A}_i \in \mathbb{R}^{n_i \times n_i}$ encodes the weighted connectivity of $\mathcal{G}_i$. A meaningful matching can be obtained between any two graphs $\mathcal{G}_i, \mathcal{G}_j$, which is denoted by the assignment matrix $\mathbf{X}_{ij} \in \{0,1\}^{n_i \times n_j}, s.t. \ \mathbf{X}_{ij}\mathbf{1} = \mathbf{1}, \mathbf{X}_{ij}^\top\mathbf{1} = \mathbf{1}$. $\mathbf{1}$ denotes a column vector whose elements are all ones.

MGM aims to mitigate wrong matchings obtained by pairwise graph matching solvers, by leveraging the namely cycle-consistency property when multiple graphs are jointly considered.

**Definition 1 Cycle-consistency [10]**. *The matching among $\mathcal{G}_1, \mathcal{G}_2, \ldots, \mathcal{G}_m$ is cycle-consistent, if*

$$\mathbf{X}_{ij} = \mathbf{X}_{ik}\mathbf{X}_{kj}, \quad \forall i,j,k \in [m] \tag{1}$$

*where $[m]$ denotes the set of graph indices from $1$ to $m$.*

We enforce cycle-consistency with our MGM method, and therefore the matching can be decomposed by matching to the universe of nodes. The matching between $\mathcal{G}_i$ and the universe of size $d$ is denoted by $\mathbf{U}_i \in \{0,1\}^{n_i \times d}$. The pairwise matching between $\mathcal{G}_i$ and $\mathcal{G}_j$ follows $\mathbf{X}_{ij} = \mathbf{U}_i\mathbf{U}_j^\top$.

Our method works with the popular formulation of pairwise graph matching, namely Koopmans-Beckmann's Quadratic Assignment Problem [42] (abbreviated as KB-QAP) for $\mathcal{G}_i, \mathcal{G}_j$:

$$\max_{\mathbf{X}_{ij}} \lambda \operatorname{tr}(\mathbf{X}_{ij}^\top \mathbf{A}_i \mathbf{X}_{ij} \mathbf{A}_j) + \operatorname{tr}(\mathbf{X}_{ij}^\top \mathbf{W}_{ij}) \quad s.t. \ \mathbf{X}_{ij} \in \{0,1\}^{n_i \times n_j}, \mathbf{X}_{ij}\mathbf{1} = \mathbf{1}, \mathbf{X}_{ij}^\top\mathbf{1} = \mathbf{1} \tag{2}$$

where $\lambda$ is the weight for the edge-to-edge similarity term, $\mathbf{A}_i, \mathbf{A}_j$ are weighted adjacency matrices of $\mathcal{G}_i, \mathcal{G}_j$, and $\mathbf{W}_{ij}$ represents the node-to-node similarity between $\mathcal{G}_i, \mathcal{G}_j$.

**Definition 2 Multi-Graph Koopmans-Beckmann's QAP**. *Multi-graph matching is formulated with KB-QAP, by summing KB-QAP objectives among all pairs of graphs:*

$$\max_{\mathbf{X}_{i,j}, i,j \in [m]} \sum_{i,j \in [m]} \left( \lambda \operatorname{tr}(\mathbf{X}_{ij}^\top \mathbf{A}_i \mathbf{X}_{ij} \mathbf{A}_j) + \operatorname{tr}(\mathbf{X}_{ij}^\top \mathbf{W}_{ij}) \right) \tag{3}$$

*where all $\mathbf{X}_{ij}$ are cycle-consistent and $i, j \in [m]$ means iterating among all combinations of $i, j$. The constraints in Eq. 3 are omitted for compact illustration.*

#### 2.1.2 The Proposed MGM Method

Our graduated assignment multi-graph matching (GA-MGM) works by iterative projection based on the Taylor expansion of MGM KB-QAP. The proposed method is summarized in Alg. 1. Our method handles partial matching and is capable to be extended on joint clustering and matching. The clustering weight matrix $\mathbb{B}$ has no effect with MGM problem and is filled with all ones.

**Computing node-wise similarity**. Firstly, node-wise similarity is computed from the inner-product of node features in graph pair $\mathcal{G}_i, \mathcal{G}_j$: $\mathbf{W}_{ij} = \operatorname{Sinkhorn}(\mathbf{F}_i\mathbf{F}_j^\top, \tau_w)$ where $\operatorname{Sinkhorn}(\mathbf{M}, \tau)$ is the popular Sinkhorn algorithm for matrix normalization [24], which will be discussed later in details.

**Algorithm 1: Graduated Assignment Multi-Graph Matching (GA-MGM)**

**Input:** Input graphs $\{\mathcal{G}_1, \mathcal{G}_2, ...\mathcal{G}_m\}$; node-wise similarity $\{\mathbf{W}_{ij}\}$; initial annealing $\tau_0$; descent factor $\gamma$; minimum $\tau_{min}$; clustering weight $\mathbb{B}$ (all $\mathbb{B}_{ij} = 1$ if clustering is not considered).

1  Randomly initialize joint matching $\{\mathbf{U}_i\}$; projector $\leftarrow$ Sinkhorn; $\tau \leftarrow \tau_0$;
2  **while** *True* **do**
3      **while** $\{\mathbf{U}_i\}$ *not converged AND #iter $\leq$ #MGMIter* **do**
4          $\forall i \in [m], \mathbf{V}_i \leftarrow \mathbf{0}$;
5          **for** $\mathcal{G}_i, \mathcal{G}_j$ *in* $\{\mathcal{G}_1, \mathcal{G}_2, ...\mathcal{G}_m\}$ **do**
6              $\mathbf{V}_i \leftarrow \mathbf{V}_i + (\lambda \mathbf{A}_i \mathbf{U}_i \mathbf{U}_j^{\top} \mathbf{A}_j \mathbf{U}_j + \mathbf{W}_{ij}\mathbf{U}_j) \times \mathbb{B}_{ij}$; # update $\mathbf{V}_i$
7          **for** $\mathcal{G}_i$ *in* $\{\mathcal{G}_1, \mathcal{G}_2, ...\mathcal{G}_m\}$ **do**
8              $\mathbf{U}_i \leftarrow \text{projector}(\mathbf{V}_i, \tau)$; # project $\mathbf{V}_i$ to (relaxed) feasible space of $\mathbf{U}_i$
9      # graduated assignment control
10     **if** projector == Sinkhorn *AND* $\tau \geq \tau_{min}$ **then**
11         $\tau \leftarrow \tau \times \gamma$;
12     **else if** projector == Sinkhorn *AND* $\tau < \tau_{min}$ **then**
13         projector $\leftarrow$ Hungarian;
14     **else**
15         **break**;

**Output:** Joint matching matrices $\{\mathbf{U}_i\}$.

**Construction of edges**. We follow [34] when constructing the edges. For the weighted adjacency matrix of $\mathcal{G}_i$, we firstly compute the Euclidean distance between every pair of keypoints: $l_{ab} = ||p_a - p_b||$ where $p_a, p_b$ are the coordinates of keypoints. The corresponding $\mathbf{A}_i[a, b]$ is computed as

$$\mathbf{A}_i[a,b] = \exp(-\frac{l_{ab}^2}{\hat{l}^2}) \tag{4}$$

where $\hat{l}$ is the median value of all $l_{ab}$. The diagonal part of $\mathbf{A}_i$ is set as zeros.

**Initialization**. Each element in $\{\mathbf{U}_i\}$ is initialized by $1/d + 10^{-3}z$, where $z \sim N(0, 1)$. In comparison, some peer methods require initialization from pairwise matching [30, 11, 13] or multi-matching [34].

**Graduated assignment**. Our method works by iteratively projecting the multi-graph KB-QAP objective function Eq. 3 to the feasible space, by graduated Sinkhorn and Hungarian assignment. For non-square matrices, minus infinite are padded to form square matrices as the common technique.

A real-valued square matrix $\mathbf{M}$ can be projected to a 0/1 assignment matrix by Hungarian algorithm [43] at $\mathcal{O}(n^3)$ time complexity, which maximizes the objective of the linear assignment problem:

$$\max_{\mathbf{X}} \text{tr}(\mathbf{X}^{\top}\mathbf{M}), \quad s.t. \quad \mathbf{X} \in \{0,1\}^{n \times n}, \mathbf{X}\mathbf{1} = \mathbf{1}, \mathbf{X}^{\top}\mathbf{1} = \mathbf{1} \tag{5}$$

where $\mathbf{X}$ is the discrete assignment matrix. A relaxed projection to doubly-stochastic matrix can be achieved by Sinkhorn algorithm with namely entropic regularization [44, 23]:

$$\max_{\mathbf{S}} \text{tr}(\mathbf{S}^{\top}\mathbf{M}) - \tau h(\mathbf{S}), \quad s.t. \quad \mathbf{S} \in [0,1]^{n \times n}, \mathbf{S}\mathbf{1} = \mathbf{1}, \mathbf{S}^{\top}\mathbf{1} = \mathbf{1} \tag{6}$$

where $\mathbf{S}$ is the doubly-stochastic matrix, $h(\mathbf{S}) = \sum_{i,j} \mathbf{S}_{ij} \log \mathbf{S}_{ij}$ is the entropic regularizer and $\tau \in (0, +\infty)$ is the regularization factor. Given any real-valued matrix $\mathbf{M}$, it is firstly processed by regularization factor $\tau$: $\mathbf{S} = \exp(\mathbf{M}/\tau)$. Then $\mathbf{S}$ is row- and column- wise normalized alternatively:

$$\mathbf{D}_r = \text{diag}(\mathbf{S}\mathbf{1}), \quad \mathbf{S} = \mathbf{D}_r^{-1}\mathbf{S}, \quad \mathbf{D}_c = \text{diag}(\mathbf{S}^{\top}\mathbf{1}), \quad \mathbf{S} = \mathbf{S}\mathbf{D}_c^{-1} \tag{7}$$

where $\text{diag}(\cdot)$ means building a diagonal matrix from the input vector. The converged $\mathbf{S}$ is the doubly-stochastic relaxation from the discrete solution solved by Hungarian algorithm. Readers are referred to [25] for the theoretical analysis on entropically regularized Sinkhorn algorithm.

The gap between Sinkhorn and Hungarian algorithm is controlled by $\tau$, which can be viewed as an annealing parameter: Sinkhorn algorithm performs closely to Hungarian if $\tau$ is small (at the

cost of slowed convergence), and the output of Sinkhorn will become smoother given larger $\tau$ [23]. Therefore, Sinkhorn with shrinking $\tau$ can be adopted to gradually project the input matrix to the feasible assignment matrix, and finally achieve the discrete matching result by Hungarian method. In Alg. 1, the shrinking speed of Sinkhorn projection is controlled by $\gamma < 1$ with $\tau \leftarrow \tau \times \gamma$ and there is a lower limit $\tau_{min}$ in our approach.

### 2.1.3 Theoretical Analysis on Solving Multi-Graph KB-QAP

The multi-graph KB-QAP objective function in Eq. 3 is abbreviated as $J$. By setting $\mathbf{X}_{ij} = \mathbf{U}_i^\top \mathbf{U}_j$, for a set of feasible multi-graph matching solution $\{\mathbf{U}_i^0\}$, $J$ can be rewritten in its Taylor series:

$$J = \sum_{i,j \in [m]} \lambda \operatorname{tr}(\mathbf{U}_j^0 \mathbf{U}_i^{0\top} \mathbf{A}_i \mathbf{U}_i^0 \mathbf{U}_j^{0\top} \mathbf{A}_j) + \operatorname{tr}(\mathbf{U}_j^0 \mathbf{U}_i^{0\top} \mathbf{W}_{ij}) + \sum_{i \in [m]} \operatorname{tr}(\mathbf{V}_i^\top (\mathbf{U}_i - \mathbf{U}_i^0)) + \dots$$

$$\text{where} \qquad \mathbf{V}_i = \left. \frac{\partial J}{\partial \mathbf{U}_i} \right|_{\mathbf{U}_i = \mathbf{U}_i^0} = \sum_{j \in [m]} \left( 2\lambda \mathbf{A}_i \mathbf{U}_i^0 \mathbf{U}_j^{0\top} \mathbf{A}_j \mathbf{U}_j^0 + \mathbf{W}_{ij} \mathbf{U}_j^0 \right) \tag{8}$$

Approximating $J$ with first-order Taylor expansion, the maximization of $J$ is equivalent to maximizing: $\sum_{i \in [m]} \operatorname{tr}(\mathbf{V}_i^\top \mathbf{U}_i)$, which is equivalent to solving $m$ independent linear assignment problems. Therefore, given initial $\mathbf{U}_i$, our GA-MGM works by considering the first-order Taylor expansion of the multi-graph KB-QAP objective and computing $\mathbf{V}_i$ by Eq. 8. The above linear assignment problems are solved via either Sinkhorn algorithm (controlled by $\tau$) [23] or Hungarian algorithm [43]. The solver to linear assignment is carefully controlled by the annealing parameter $\tau$ of Sinkhorn algorithm and finally Hungarian algorithm, ensuring $\mathbf{U}_i$ gradually converges to a high-quality discrete solution.

## 2.2 Graduated Assignment for Multi-Graph Matching and Clustering

### 2.2.1 The Multi-Graph Matching and Clustering Problem

We further consider the more general MGMC setting, where the set of all graphs is a mixture of graphs from $k$ categories $\mathcal{C}_1, \mathcal{C}_2, \dots, \mathcal{C}_k$ and $|\mathcal{C}_1 \cup \mathcal{C}_2 \cup \cdots \mathcal{C}_k| = m$, $\mathcal{C}_i \cap \mathcal{C}_j = \emptyset$ for all $i, j \in [k]$. For MGMC problem, the method needs to divide all graphs to $k$ clusters, and for every two graphs from the same cluster, a node-to-node correspondence is computed for the matching problem.

The objective of MGMC problem is formulated by slight modification from MGM KB-QAP Eq. 3:

$$\max_{\mathbf{X}_{ij}, i, j \in [m]} \sum_{i, j \in [m]} \mathbb{C}_{ij} \left( \lambda \operatorname{tr}(\mathbf{X}_{ij}^\top \mathbf{A}_i \mathbf{X}_{ij} \mathbf{A}_j) + \operatorname{tr}(\mathbf{X}_{ij}^\top \mathbf{W}_{ij}) \right) \tag{9}$$

where $\mathbb{C} \in \{0, 1\}^{m \times m}$ is the clustering matrix. If $\mathcal{G}_i, \mathcal{G}_j$ are in the same category then $\mathbb{C}_{ij} = 1$ else $\mathbb{C}_{ij} = 0$. We represent clustering-related variables by blackboard bold letters in this paper.

### 2.2.2 The Proposed GA-MGMC Method

Based on GA-MGM pipeline, our proposed graduated assignment multi-graph matching and clustering (GA-MGMC) method solves the clustering problem and matching problem simultaneously. Motivated by the intuitive idea that more precise multi-graph matching will improve the clustering accuracy and vice versa, GA-MGMC predicts clustering and multi-graph matching alternatively with gradually increased clustering confidence until convergence. GA-MGMC is summarized in Alg. 2.

**Matching-based clustering**. The key challenge of multi-graph clustering is finding a reasonable measurement for graph-wise similarity, after which the common spectral clustering technique can be applied to discover clusters. We tackle this problem by proposing a matching-based graph-wise similarity measure for clustering.

Given a batch of graphs from multiple categories, a multi-graph matching relationship can be achieved by out-of-box solvers e.g. our proposed GA-MGM. For graphs in the same category, there should be a higher agreement in their structural and node-wise alignment, compared to graphs from different categories. Therefore, the matching information among graphs can be adopted as the similarity measure. For $\mathcal{G}_i, \mathcal{G}_j$, their similarity is computed from their node-wise and structural agreement:

$$\mathbb{A}_{ij} = \underbrace{\lambda_c \exp(-||\mathbf{X}_{ij}^\top \mathbf{A}_i \mathbf{X}_{ij} - \mathbf{A}_j||)}_{\text{structural agreement}} + \underbrace{\operatorname{tr}(\mathbf{X}_{ij}^\top \mathbf{W}_{ij})}_{\text{node-wise agreement}} \tag{10}$$

**Algorithm 2: Graduated Assignment Multi-Graph Matching and Clustering (GA-MGMC)**

**Input:** Input graphs $\{\mathcal{G}_1, \mathcal{G}_2, ...\mathcal{G}_m\}$; node-wise similarity $\{\mathbf{W}_{ij}\}$; list of clustering weight $\{\beta\}$.

1  All $\mathbb{B}_{ij} \leftarrow 1$; # initialize clustering weight matrix as all ones
2  **for** $\beta$ *in* $\{\beta\}$ **do**
3      **while** $\{\mathbf{U}_i\}$, $\mathbb{C}$ *not converged AND #iter $\leq$ #MGMCIter* **do**
4          $\{\mathbf{U}_i\} \leftarrow$ GA-MGM$(\{\mathcal{G}_1, \mathcal{G}_2, ...\mathcal{G}_m\}, \{\mathbf{W}_{ij}\}, \mathbb{B})$; # multi-graph matching
5          **for** $\mathcal{G}_i, \mathcal{G}_j$ *in* $\{\mathcal{G}_1, \mathcal{G}_2, ...\mathcal{G}_m\}$ **do**
6              build $\mathbf{X}_{ij}$ from $\mathbf{U}$;
7              $\mathbb{A}_{ij} \leftarrow \exp(-||\mathbf{X}_{ij}^\top \mathbf{A}_i \mathbf{X}_{ij} - \mathbf{A}_j||) + \mathrm{tr}(\mathbf{X}_{ij}^\top \mathbf{W}_{ij})$ # graph-to-graph similarity
8          $\mathbb{C} \leftarrow$ spectral clustering with k-means++ on $\mathbb{A}$;
9          $\mathbb{B} \leftarrow \mathbb{C} \times (1 - \beta) + \beta$; # clustering weight

**Output:** Joint matching matrices $\{\mathbf{U}_i\}$; Joint clustering matrix $\mathbb{C}$.

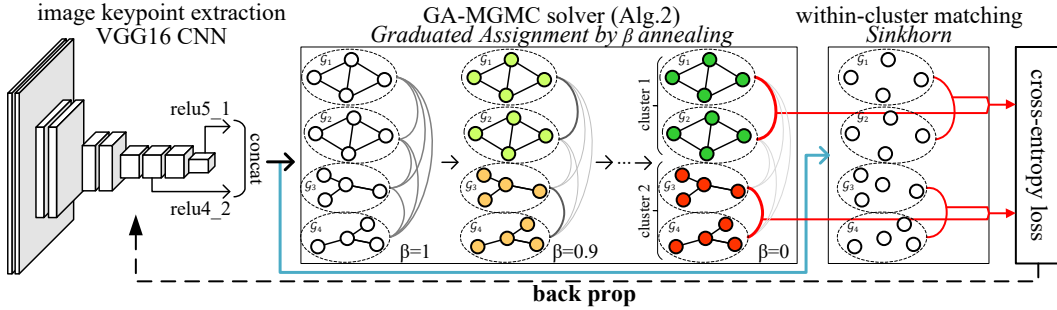

Figure 1: Our unsupervised MGMC learning pipeline with two clusters (each contains two graphs) for illustration. As Alg. 2 iterates with decreasing annealing parameter $\beta$, the two-graph matching confidence increases (the darker the higher) and so for the clustering confidence as indicated by the brightened node color from white to green/red. In parallel, CNN and Sinkhorn net can form a two-graph matching network (connected by the blue arrows) whose samples are those within each cluster determined by Alg. 2 and its cross-entropy loss is computed based on the input matching from the above matching network and the pseudo matching labels predicted by Alg. 2. As such, the VGG16 layers can be trained via back-propagation along the flow in black.

where the first entry measures the similarity of aligned adjacency matrices and the second entry encodes the agreement on node-wise alignment. The weighting factor $\lambda$ in MGM is also considered here. Spectral clustering with k-means++ [45] is further performed on $\mathbb{A}$ as the common technique. Based on Eq. 10, for intra-cluster graphs $\mathcal{G}_i, \mathcal{G}_j$, a more accurate $\mathbf{X}_{ij}$ will result in larger $\mathbb{A}_{ij}$, therefore increased matching accuracy will lead to more accurate clustering.

**Clustering-aware matching**. Following the clustering-aware MGM KB-QAP formulation in Eq. 9, the clustering weight $\mathbb{B}_{ij}$ is multiplied to the projection step of GA-MGM (L6 in Alg. 1). If we assign $\mathbb{B} = \mathbb{C}$, the projection step meets the objective function in Eq. 9, assuming 100% clustering accuracy. In realistic conditions with non-optimal clustering, we further apply an annealing parameter $\beta$ for the clustering weight matrix $\mathbb{B}$: $\mathbb{B}_{ij} = \begin{cases} 1 & \text{if } \mathbb{C}_{ij} = 1 \\ \beta & \text{if } \mathbb{C}_{ij} = 0 \end{cases}$ which is equivalent to $\mathbb{B} = \mathbb{C} \times (1 - \beta) + \beta$. The annealing parameter $\beta \in [0, 1]$ can be viewed as the confidence of clustering result, and joint clustering and matching can be achieved by gradually declining $\beta$ from 1 to 0.

### 2.3 Graduated Assignment Network with Unsupervised Learning

We name the unsupervised learning pipeline as graduated assignment neural network (GANN) for both MGM and MGMC. An overview of our unsupervised learning pipeline GANN-MGMC is shown in Fig. 1, and MGM learning is obtained by replacing GA-MGMC with GA-MGM and ignoring clustering. Based on previously discussed graduated assignment matching algorithms, the image feature extractor VGG16 is learned by pseudo labels predicted by GA-MGM or GA-MGMC, with gradient propagated with differentiable Sinkhorn layer. For MGM problem, GA-MGM in Alg. 1 solves a cycle-consistent matching relation, which is further adopted to supervise the learning on

Table 1: Parameter configurations to reproduce our reported results in this paper.

| param | Willow Object Class | | CUB2011 | | description |
|---|---|---|---|---|---|
| | MGM | MGMC | MGM | MGMC | |
| lr | $10^{-3}$ | $10^{-3}$ | $10^{-3}$ | $10^{-4}$ | learning rate |
| lr-steps | {200,1000} | {100,500} | {2000} | {1000} | lr /= 10 at these number of iterations |
| $\lambda$ | 0.5 | 0.5 | 0.05 | 0.05 | weight of the quadratic term in matching |
| $\lambda_c$ | 1 | 1 | 0.1 | 0.1 | weight of the structural term in clustering |
| $\tau_w$ | 0.05 | 0.05 | 0.05 | 0.05 | Sinkhorn regularization for pairwise matching |
| $\{\beta\}$ | - | {1, 0.9, 0} | - | {1, 0.9} | the list of clustering weight values |
| $\tau_0$ | 0.1 | {0.1, 0.1, 0.1} | 0.05 | {0.05, 0.05} | init value of Sinkhorn regularization for GA-MGM |
| $\tau_{min}$ | $10^{-2}$ | $\{10^{-2}, 10^{-2}, 10^{-3}\}$ | $10^{-3}$ | $\{10^{-2}, 10^{-2}\}$ | lower limit of Sinkhorn regularization for GA-MGM |
| $\gamma$ | 0.8 | 0.8 | 0.8 | 0.8 | shrinking factor of tau in GA-MGM |
| #SKIter | 10 | 10 | 100 | 100 | max iterations of Sinkhorn |
| #MGMIter | 1000 | 1000 | 500 | 500 | max iterations in the innerloop of GA-MGM (Alg. 1) |
| #MGMCIter | - | 10 | - | 10 | max iterations in the loop of GA-MGMC (Alg. 2) |

VGG16 weights by minimizing the cross-entropy between MGM and pairwise matching:

$$\mathcal{L} = \sum_{i,j \in [m]} \mathrm{BCE}(\mathbf{X}_{ij}, \mathbf{W}_{ij}) \tag{11}$$

where BCE abbreviates binary cross-entropy, which is adopted by deep graph matching methods [18, 36] for supervised learning. $\bar{\mathbf{X}}_{ij}$ is cycle-consistent MGM result from GA-MGM as the pseudo label and $\mathbf{W}_{ij} = \mathrm{Sinkhorn}(\mathbf{F}_i \mathbf{F}_j^\top, \tau_w)$ is pairwise matching composed from individual node features.

For MGMC problem, GA-MGMC is adopted as the clustering and matching solver, which predicts additionally $\mathbb{C}$ as the clustering matrix. The loss involves a clustering indicator $I(i,j)$:

$$\mathcal{L} = \sum_{i,j \in [m]} I(i,j) \, \mathrm{BCE}(\mathbf{X}_{ij}, \mathbf{W}_{ij}) \tag{12}$$

where $I(i,j) = \mathbb{C}_{ij}$ under unsupervised setting. It is worth noting that there may exist misclassified graphs in the predicted $\mathbb{C}$, so that during training, Eq. 12 will probably involve meaningless matching between graphs from different ground truth categories, yielding challenges for learning. However, our unsupervised scheme overcomes this issue by improving matching and clustering simultaneously with graduated assignment, empirically outperforming existing learning-free peer-methods.

## 3 Experiments

**Evaluation Protocol** is built on two real-world graph matching benchmarks: Willow Object Class [16] dataset and CUB2011 dataset [46]. The VGG16 [35] backbone is initialized by ImageNet [47] and `relu4_2`, `relu5_1` features are concatenated following previous works [18, 36]. Our evaluation protocol is built in line with deep graph matching peer methods [18, 37], and we use the matlab code released by [20] for MGMC peer methods upon the authors' approval. We name our learning-free methods by graduated assignment (GA-MGM and GA-MGMC), and unsupervised learning methods by graduated assignment neural network (GANN-MGM and GANN-MGMC).

We implement Alg. 1 and 2 with GPU parallelization. Our parameter configurations are listed in Tab. 1. Experiments are conducted on our Linux workstation with Xeon-3175X@3.10GHz, RTX8000, and 128GB memory.

Given one predicted assignment matrix $\mathbf{X}$ and its ground truth $\mathbf{X}^{gt}$, precision $= \frac{\mathrm{tr}(\mathbf{X}^\top \mathbf{X}^{gt})}{\mathrm{sum}(\mathbf{X})}$, recall $= \frac{\mathrm{tr}(\mathbf{X}^\top \mathbf{X}^{gt})}{\mathrm{sum}(\mathbf{X}^{gt})}$ and the corresponding f1-score are considered for matching accuracy. Mean and STD are reported from all possible pairs of graphs. Note that precision = recall = f1 if there exists no outliers, and we denote this metric as "matching accuracy" inline with previous works [18, 36, 37, 20].

For the MGMC task, matching accuracy is evaluated with intra-cluster graphs. The following clustering metrics are also considered: **1) Clustering Purity (CP)** [48] where $\mathcal{C}_i$ represent the predicted cluster $i$ and $\mathcal{C}_j$ is ground truth cluster $j$: CP $= \frac{1}{m} \sum_{i=1}^{k} \max_{j \in \{1,\dots,k\}} |\mathcal{C}_i \cap \mathcal{C}_j^{gt}|$; **2) Rand Index (RI)** [49] computed by the number of graphs predicted in the same cluster with the same label $n_{11}$ and the number of graphs predicted in separate clusters and with different labels $n_{00}$, normalized

Table 2: Matching accuracy with both learning-free MGM methods and supervised learning peer methods on Willow dataset (50 tests). Compared results are quoted from the original papers.

| method | learning | car | duck | face | mbike | wbottle |
|---|---|---|---|---|---|---|
| MatchLift [13] | free | 0.665 | 0.554 | 0.931 | 0.296 | 0.700 |
| MatchALS [30] | free | 0.629 | 0.525 | 0.934 | 0.310 | 0.669 |
| MSIM [12] | free | 0.750 | 0.732 | 0.937 | 0.653 | 0.814 |
| HiPPI [34] | free | 0.740 | 0.880 | **1.000** | 0.840 | **0.950** |
| MGM-Floyd [27] | free | **0.850** | 0.793 | **1.000** | 0.843 | 0.931 |
| GA-MGM (ours) | free | 0.746±0.153 | **0.900±0.106** | 0.997±0.021 | **0.892±0.139** | 0.937±0.072 |
| HARG-SSVM [16] | supervised | 0.584 | 0.552 | 0.912 | 0.444 | 0.666 |
| GMN [17] | supervised | 0.743 | 0.828 | 0.993 | 0.714 | 0.767 |
| PCA-GM [18] | supervised | 0.840 | 0.935 | **1.000** | 0.767 | 0.969 |
| DGMC [37] | supervised | 0.903 | 0.890 | **1.000** | 0.921 | 0.971 |
| CIE [36] | supervised | 0.822 | 0.812 | **1.000** | 0.900 | 0.976 |
| NMGM [41] | supervised | **0.973** | 0.854 | **1.000** | 0.860 | 0.977 |
| GANN-MGM (ours) | unsup. | 0.964±0.058 | **0.949±0.057** | **1.000±0.000** | **1.000±0.000** | **0.978±0.035** |

Table 3: Multiple graph matching and clustering evaluation (with inference time) on Willow dataset.

| method | learning | 8 Cars, 8 Ducks, 8 Motorbikes | | | | | 40 Cars, 50 Ducks, 40 Motorbikes | | | | |
|---|---|---|---|---|---|---|---|---|---|---|---|
| | | CP | RI | CA | MA | time (s) | CP | RI | CA | MA | time (s) |
| RRWM [2] | free | 0.879 | 0.871 | 0.815 | 0.748 | **0.4** | 0.962 | 0.949 | 0.926 | 0.751 | **8.8** |
| CAO-C [10] | free | 0.908 | 0.903 | 0.860 | 0.878 | 3.3 | 0.971 | 0.960 | **0.956** | 0.906 | 1051.5 |
| CAO-PC [10] | free | 0.887 | 0.883 | 0.831 | 0.870 | 1.8 | 0.971 | 0.960 | **0.956** | 0.886 | 184.0 |
| DPMC [20] | free | 0.931 | 0.923 | 0.890 | 0.872 | 1.2 | 0.969 | 0.959 | 0.948 | **0.941** | 97.5 |
| GA-MGMC (ours) | free | 0.921 | 0.905 | 0.893 | 0.653 | 10.6 | 0.890 | 0.871 | 0.850 | 0.669 | 107.8 |
| GANN-MGMC (ours) | unsup. | **0.976** | **0.970** | **0.963** | **0.896** | 5.2 | **0.974** | **0.968** | **0.956** | 0.906 | 80.7 |

by the total number of graph pairs $n$: $\text{RI} = \frac{n_{11}+n_{00}}{n}$; **3) Clustering Accuracy (CA)** [20] where $\mathcal{A}, \mathcal{B}, ...$ are ground truth clusters and $\mathcal{A}', \mathcal{B}', ...$ are predicted clusters and $k$ is the number of clusters: $\text{CA} = 1 - \frac{1}{k}\left(\sum_{\mathcal{A}}\sum_{\mathcal{A}'\neq\mathcal{B}'}\frac{|\mathcal{A}'\cap\mathcal{A}||\mathcal{B}'\cap\mathcal{A}|}{|\mathcal{A}||\mathcal{A}|} + \sum_{\mathcal{A}'}\sum_{\mathcal{A}\neq\mathcal{B}}\frac{|\mathcal{A}'\cap\mathcal{A}||\mathcal{A}'\cap\mathcal{B}|}{|\mathcal{A}||\mathcal{B}|}\right)$.

**Willow Object Class** [16] contains 304 images from 5 categories, collected from Caltech-256 [50] (face, duck and winebottle) and Pascal VOC 2007 [51] (car and motorbike). Each image contains one object and is labeled with 10 common semantic keypoints with no outliers. Both MGM and MGMC problems are evaluated. For both problems, our models are learned with 8 graphs randomly drawn per category. Based on unsupervised learning, the whole dataset is utilized for both training and testing.

Our evaluation protocol on MGM is in line with [41, 27], where both learning-free and learning-based methods are compared, as shown in Tab. 2. Most compared learning graph matching methods are two-graph matching since there is little effort on learning MGM except NMGM [41]. Among learning-free methods, our GA-MGM performs comparatively with state-of-the-art MGM solvers. Most importantly, under the learning setting, our unsupervised learning GANN-MGM surpasses all supervised two-graph matching learning peer methods and best performs on 4 out of 5 categories.

The evaluation on MGMC problem is considered in line with [20], where matching and clustering are simultaneously solved for graphs from 3 Willow categories (car, duck, motorbike). Since there exists no learning-based peer method for MGMC problem, only learning-free algorithms are considered. Apart from the dedicated DPMC solver [20] for MGMC problem, popular methods on two-graph matching [2] and multi-graph matching [10] are also considered, where matching is firstly solved among all graphs, followed by spectral clustering. Clustering metrics CP, RI, CA, and intra ground-truth cluster matching accuracy (MA) are reported, as shown in Tab. 3. Both smaller-scaled (24 graphs) and larger-scaled (130 graphs) MGMC problems are experimented, and our GANN-MGMC best performs on the smaller-scaled problem and scales up soundly. For the MGMC task, accurate matching is actually meaningless for misclassified graphs. This statement can be addressed by unsupervised learning, as learning-free GA-MGMC achieves accurate clustering but is not as comparative in matching, however, both matching and clustering performances can be improved with supervision from GA-MGMC. Our GANN-MGMC is relatively slow on the smaller-scaled problem, but runs comparatively fast with DPMC [20] on the larger-scaled problem, probably because the overhead on VGG16 is more significant on small problems. We also discover unsupervised learning improves the convergence speed of our graduated assignment method.

**CUB2011** dataset [46] includes 11,788 bird images from 200 categories with closely 50%/50% split of training/testing images for each category, and each bird is labeled with a partial of 15 keypoints.

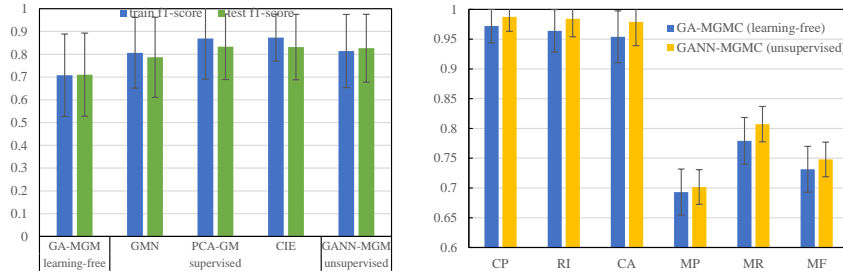

Figure 2: Mean and STD on MGM (left) and MGMC (right) on CUB2011 dataset.

CUB2011 is more challenging compared to Willow Object because it is of larger size and need to deal with partial matching. Both MGM and MGMC problems are evaluated with CUB2011.

For the MGM problem, as the learning-free version of our method is comparative with the recent learning-free peer MGM methods [12, 27, 34], we mainly compare with the available supervised deep learning methods (only for two graphs). For our method, during training we randomly draw 8 graphs per category for cost-efficiency, and in evaluation all graphs from the same category are matched jointly. For the MGMC task, evaluation is conducted on 10 Horned Grebe, 10 Baird Sparrow, and 10 Caspian Tern for 50 tests. This MGMC problem is more challenging than the Willow dataset, because all involved images belong to different subcategories of birds and are more difficult to categorize.

We follow the train/test split of the original dataset, and our unsupervised learning method is learned on the training set. Fig. 2 includes results on both MGM and MGMC. For the MGM task, our GANN-MGM performs comparatively on the testing set against state-of-the-art PCA-GM [18] and CIE [36] which are trained by ground truth supervision. Also note that our unsupervised learning scheme generalizes soundly from training samples to unseen testing samples, while the supervised learning methods suffer from overfitting on training data. For the MGMC problem, clustering metrics and matching precision (MP), recall (MR) and f1-score (MF) are reported on the testing set. Unsupervised learning helps to improve all matching and clustering metrics compared to the learning-free version.

**Further study: Validation of graduated assignment on MGMC problem**. In Fig. 3, we plot the clustering and matching metrics when the learning-free GA-MGMC model deals with the MGMC problem on Willow Object Class dataset (40 Cars, 50 Ducks, 40 Motorbikes, in line with right half of Tab. 3). As shown in Fig. 3, the algorithm starts with no clustering information, resulting in a low-quality matching at the first iteration, with which a moderately good clustering is achieved. With decreased clustering weight $\beta$, more accurate clustering again improves matching accuracy and vice versa, fi-

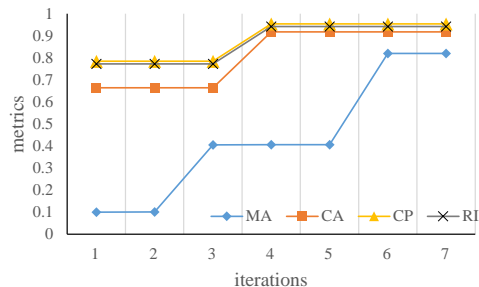

Figure 3: Study on the effectiveness of gradually decreasing the clustering weight $\beta$ from 1 to 0 for the MGMC problem.

nally reaching a satisfying matching and clustering result. If both matching and clustering do not change, it means that they have converged and $\beta$ is declined at next iteration. When $\beta$ declines, the matching accuracy improves because more confidence is counted on the clustering result, and the clustering accuracy also improves when more accurate matching is obtained. Therefore, our GA-MGMC improves matching and clustering simultaneously with graduated assignment.

## 4  Conclusions

We have presented a joint graph matching and clustering method, whose matching results have also been shown can serve as pseudo ground truth to train a two-graph matching network. To our best knowledge, this is the first deep learning based approach to such a practical problem. Promising results are obtained showing the feasibility and advantage of introducing deep networks.

## Broader Impact

*a) Who may benefit from this research.* In this paper, a graduated assignment approach is proposed for two challenging real-world problems – multi-graph matching (MGM) and multi-graph matching and clustering (MGMC), together with an unsupervised learning scheme for both problems. The wide range of applications in pattern recognition and data mining may benefit from this research, as unsupervised learning with comparative performance with supervised learning is usually welcomed for real-world applications.

*b) Who may be put at disadvantage from this research.* Multi-graph matching and multi-graph matching and clustering techniques may be applied to some intensive areas, e.g. analyzing surveillance sequences, which improves security but potentially puts citizens' privacy at risk. Therefore, the potentially affected people should be well informed before any deployment of our technique in such intensive areas.

*c) What are the consequences of failure of the system.* A failure of our system may result in totally meaningless matching and clustering results, further affecting the algorithm for downstream tasks. It is worth noting that unsupervised learning approach may overfit on small-scaled training data, and people should carefully validate the efficacy of the system before deploying to real-world scenes.

*d) Whether the task/method leverages biases in the data.* Although unsupervised learning may overfit on biased training data, we think our unsupervised learning approach helps to mitigate such issue because its data collection is cheaper compared to supervised learning, making it easier to build an unbiased training set. Furthermore, within the scope of our experiments, our unsupervised learning seems to not leverage the bias in data, as our model performs comparatively on both seen an unseen data during training, as shown in the top of Fig. 2.

## Acknowledgments and Disclosure of Funding

This work was partially supported by National Key Research and Development Program of China 2020AAA0107600, and NSFC (61972250, U19B2035, 72061127003), and CCF-Tencent Open Fund RAGR20200113 and Tencent AI Lab Rhino-Bird Visiting Scholars Program. The author Runzhong Wang is also sponsored by Wen-Tsun Wu Honorary Doctoral Scholarship, AI Institute, Shanghai Jiao Tong University.

## Footnotes

*Junchi Yan is the corresponding author.

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
