[Supplementary Material]

# Supplementary Material

## A  Detailed derivation on Eq. 7

For detailed derivation of Eq. 7, firstly we write $J$ in the form of element-wise computation. Here $\mathbf{U}_i[a, b]$ denotes the $(a, b)$ element of matrix $\mathbf{U}_i$.

$$
\begin{aligned}
J &= \sum_{i,j \in [m]} \left( \lambda \operatorname{tr}(\mathbf{U}_j \mathbf{U}_i^\top \mathbf{A}_i \mathbf{U}_i \mathbf{U}_j^\top \mathbf{A}_j) + \operatorname{tr}(\mathbf{U}_j \mathbf{U}_i^\top \mathbf{W}_{ij}) \right) \\
&= \sum_{i,j \in [m]} \left( \lambda \operatorname{tr}(\mathbf{U}_i^\top \mathbf{A}_i \mathbf{U}_i \mathbf{U}_j^\top \mathbf{A}_j \mathbf{U}_j) + \operatorname{tr}(\mathbf{U}_i^\top \mathbf{W}_{ij} \mathbf{U}_j) \right) \\
&= \sum_{i,j \in [m]} \left( \lambda \sum_{y=1}^{d} \sum_{a=1}^{n_i} \sum_{b=1}^{n_i} \sum_{z=1}^{d} \sum_{p=1}^{n_j} \sum_{q=1}^{n_j} \mathbf{U}_i[a, y] \mathbf{A}_i[a, b] \mathbf{U}_i[b, z] \mathbf{U}_j[p, z] \mathbf{A}_j[p, q] \mathbf{U}_j[q, y] \right. \\
&\quad \left. + \sum_{y=1}^{d} \sum_{a=1}^{n_i} \sum_{p=1}^{n_j} \mathbf{U}_i[a, y] \mathbf{W}_{ij}[a, p] \mathbf{U}_j[p, y] \right)
\end{aligned}
\tag{12}
$$

computing the partial derivative with respect to $\mathbf{U}_i[a, y]$:

$$
\begin{aligned}
\frac{\partial J}{\partial \mathbf{U}_i[a, y]} = \sum_{j \in [m]} \left( \lambda \sum_{b=1}^{n_i} \sum_{z=1}^{d} \sum_{p=1}^{n_j} \sum_{q=1}^{n_j} \mathbf{A}_i[a, b] \mathbf{U}_i[b, z] \mathbf{U}_j[p, z] \mathbf{A}_j[p, q] \mathbf{U}_j[q, y] \right. \\
\left. + \sum_{p=1}^{n_j} \mathbf{W}_{ij}[a, p] \mathbf{U}_j[p, y] \right)
\end{aligned}
\tag{13}
$$

and it can be equivalently written in the matrix form:

$$
\frac{\partial J}{\partial \mathbf{U}_i} = \sum_{j \in [m]} \left( \lambda \mathbf{A}_i \mathbf{U}_i \mathbf{U}_j^\top \mathbf{A}_j \mathbf{U}_j + \mathbf{W}_{ij} \mathbf{U}_j \right)
\tag{14}
$$

By substituting $\mathbf{U}_i = \mathbf{U}_i^0$, we will get Eq. 7.

## B  Implementation details and discussions

### B.1  Compact matrix form implementation

The multi-graph KB-QAP objective in Eq. 3 can be equivalently written in a compact matrix form, whereby the computational efficiency can be easily benefited with GPU parallelization. Inspired by [32], the matching objective in Eq. 3 can be written as:

$$
\max_{\mathbf{U}} \ \lambda \operatorname{tr}(\mathbf{U}\mathbf{U}^\top \mathbf{A} \mathbf{U} \mathbf{U}^\top \mathbf{A}) + \operatorname{tr}(\mathbf{U}\mathbf{U}^\top \mathbf{W})
\tag{15}
$$

where $\mathbf{U}$ is the joint matching matrix by stacking all $\mathbf{U}_i$ at their first dimension. $\mathbf{A}$ is the joint adjacency matrix by placing $\mathbf{A}_i$ at its diagonal, and $\mathbf{W}$ is the joint node-to-node similarity matrix:

$$
\mathbf{U} = \begin{pmatrix} \mathbf{U}_0 \\ \vdots \\ \mathbf{U}_m \end{pmatrix}, \quad \mathbf{A} = \begin{pmatrix} \mathbf{A}_0 & 0 & \cdots & 0 \\ 0 & \mathbf{A}_1 & \cdots & 0 \\ \vdots & \vdots & \ddots & \vdots \\ 0 & 0 & \cdots & \mathbf{A}_m \end{pmatrix}, \quad \mathbf{W} = \begin{pmatrix} \mathbf{W}_{00} & \cdots & \mathbf{W}_{0m} \\ \vdots & \ddots & \vdots \\ \mathbf{W}_{m0} & \cdots & \mathbf{W}_{mm} \end{pmatrix}
$$

The update step of GA-MGM (L6 of Alg. 1) can be replaced with

$$
\mathbf{V} \leftarrow \lambda \mathbf{A} \mathbf{U} \mathbf{U}^\top \mathbf{A} \mathbf{U} + \mathbf{W} \mathbf{U}
\tag{16}
$$

The clustering weight can also be fused with this compact matrix form, by modifying Eq. 8 as

$$
\max_{\mathbf{U}} \ \lambda \operatorname{tr}(\mathbf{U}\mathbf{U}^\top \mathbf{A} (\mathbf{U}\mathbf{U}^\top \circ \mathbb{M}) \mathbf{A}) + \operatorname{tr}(\mathbf{U}\mathbf{U}^\top \mathbf{W})
\tag{17}
$$

Table 3: The averaged inference time of learning-free GA-MGM and GA-MGMC w/ or w/o the compact matrix form, on the Willow dataset (in line with Tab. 1 and the right half of Tab. 2).

|  | compact matrix | inference time (s) |
|---|---|---|
| GA-MGM | ✓ | **1.1** |
|  | × | 345.6 |
| GA-MGMC | ✓ | **107.2** |
|  | × | 252.0 |

where $\circ$ denotes element-wise multiplication, and $\mathbb{M}_{n_i:n_{(i+1)},n_j:n_{(j+1)}} = \mathbb{C}_{ij}$ introduces the clustering information to the multi-cluster multi-graph matching objective. Therefore, considering additionally clustering, the update step in GA-MGMC can be replaced with

$$\mathbf{V} \leftarrow \lambda \mathbf{A}(\mathbf{U}\mathbf{U}^\top \circ \mathbb{M})\mathbf{A}\mathbf{U} + (\mathbf{W} \circ \mathbb{M})\mathbf{U} \qquad (18)$$

where $\mathbb{M}_{n_i:n_{(i+1)},n_j:n_{(j+1)}} = \mathbb{B}_{ij}$ encodes the clustering weight. As shown in Tab. 3, we empirically discover that significant acceleration could be achieved with this compact matrix form, which is probably due to the low efficiency of the built-in iterations in Python and PyTorch automatically leverages the SIMD parallelization of GPU with the compact matrix form. The speedup in MGM problem is more significant, mainly because the Face dataset contains 108 images and is much larger than other categories. The speedup with GPU becomes more significant given a larger problem size.

## B.2 Initialization

**Initialization of matching.** The initialization of $\{\mathbf{U}_i\}$ (i.e. $\mathbf{U}$ with the compact matrix form) is required, as the Taylor expansion requires a set of $\{\mathbf{U}_i^0\}$ to compute the partial derivative of the first order series. Our graduated assignment method is insensitive to the initialization and can be randomly initialized, while in comparison, the line of post-processing based multi-graph matching methods requires precise initialization by either two-graph matching [28, 11, 13] or multi-graph matching [32]. In our implementation, we adopt the initialization strategy introduced in the main paper: for all $\mathbf{U}_i \in \{\mathbf{U}_i\}$, each element is initialized by $1/d + 10^{-3}z$, where $d$ is the size of universe (see discussions following Definition 1) and $z \sim N(0,1)$ is random variable from uniform distribution.

We experiment with different initialization configurations and other initialization techniques, e.g. initialize by spectral multi-matching [11], which will be discussed in Sec. E.2. Since our method is not sensitive to initialization, we adopt random initialization for its cost-efficiency.

**Initialization of projectors.** For the MGM problem, our GA-MGM introduced in Alg. 1 firstly works with coarse linear assignment solvers, i.e. Sinkhorn method with large $\tau$, then gradually converge to a high-quality discrete solution with shrinking $\tau$, and finally the discrete Hungarian algorithm. Initializing the projector in Alg. 1 with coarse Sinkhorn method works fine with the MGM problem. For the MGMC problem, where GA-MGMC (Alg. 2) repeatedly calls GA-MGM (Alg. 1) in its loop, a more cost-efficient strategy can be adopted to initialize the projector. In implementation, for each distinct value of $\beta$, the projector is initialized by Sinkhorn with a large $\tau$ when GA-MGM is called for the first time. For later iterations with the same $\beta$, the projector is initialized with the Hungarian algorithm, because only relatively small changes will occur in the clustering result, and the corresponding matching result should not change violently. Projection with the Hungarian algorithm is adequate for such circumstances. We empirically find such an initialization technique improves the inference speed and the stability of our MGMC approach.

## B.3 Construction of edges

We follow [32] when constructing the edge weights for multi-graph matching. For the weighted adjacency matrix of $\mathcal{G}_i$, we firstly compute the length between every pair of nodes: $l_{ab} = ||p_a - p_b||$ where $p_a, p_b$ are the $(x,y)$ coordinates of keypoints in images. The corresponding $\mathbf{A}_i[a,b]$ is computed as

$$\mathbf{A}_i[a,b] = \exp(-\frac{l_{ab}^2}{\sigma \, \hat{l}^2}) \qquad (19)$$

Table 4: Parameter configurations to reproduce our reported results in this paper.

| param | Willow Object Class | | CUB2011 | | description |
|---|---|---|---|---|---|
| | MGM | MGMC | MGM | MGMC | |
| lr | $10^{-3}$ | $10^{-3}$ | $10^{-3}$ | $10^{-4}$ | learning rate |
| lr-steps | {200,1000} | {100,500} | {2000} | {1000} | lr /= 10 at these number of iterations |
| $\lambda$ | 1 | 1 | 0.1 | 0.1 | weight of the quadratic term |
| $\tau_w$ | 0.05 | 0.05 | 0.05 | 0.05 | Sinkhorn regularization for pairwise matching |
| $\{\beta\}$ | - | {1, 0.9, 0} | - | {1, 0.9} | the list of clustering weight values |
| $\tau_0$ | 0.1 | {0.1, 0.1, 0.1} | 0.05 | {0.05, 0.05} | init value of Sinkhorn regularization for GA-MGM |
| $\tau_{min}$ | $10^{-2}$ | $\{10^{-2}, 10^{-2}, 10^{-3}\}$ | $10^{-3}$ | $\{10^{-2}, 10^{-2}\}$ | lower limit of Sinkhorn regularization for GA-MGM |
| $\gamma$ | 0.8 | 0.8 | 0.8 | 0.8 | shrinking factor of tau in GA-MGM |
| #SKIter | 10 | 10 | 100 | 100 | max iterations of Sinkhorn |
| #MGMIter | 1000 | 1000 | 500 | 500 | max iterations in the innerloop of GA-MGM (Alg. 1) |
| #MGMCIter | - | 10 | - | 10 | max iterations in the loop of GA-MGMC (Alg. 2) |
| $\sigma$ | 1 | 1 | 1 | 1 | the scaling factor of edges |

where $\hat{l}$ is the median value of all $l_{ab}$ and $\sigma$ is the scaling factor. The diagonal part of $\mathbf{A}_i$ is set as zeros. We empirically set $\sigma = 1$ in all experiments.

## B.4 Parameter setting

In this section we firstly present the set of parameters to reproduce the reported results, then some empirical experiences in tuning the parameters are discussed. The detailed configuration of our model parameters is listed in Tab. 4, which are tuned by greedy search on the training dataset (discussed in the following in details).

When determining the parameters, we adopt a greedy strategy because there are too many parameters. We first select the graduated assignment related parameters under the learning-free setting then the learning-related parameters. The iteration numbers #SKIter, #MGMIter, #MGMCIter are set to be large enough to ensure the convergence under most situations. $\tau_w$ follows the configuration in previous deep graph matching networks [17, 39]. $\tau_0$ is searched with $\times 2$ interval, and $\tau_{min}$ is searched with $\times 10$ interval. As validated by the experiment in Sec. E.1 and Fig. 4 that moderately good clustering can be achieved with less accurate matching, we set larger values for $\tau_{min}$ in early loops for GA-MGMC. We test different combinations of $\beta$ values including $\{1, 0.9, 0.8, 0.5, 0\}$, and select the best-performing configuration. $\gamma$ is searched at 0.05 interval, and we find $\gamma = 0.8$ best balances speed and accuracy. The quadratic weight $\lambda$ is searched at $\times 10$ interval, and $\gamma = 1, \gamma = 0.1$ seems suitable choices for Willow dataset and CUB2011 dataset, respectively. The scaling factor of edges $\sigma$ is initialized set as 1 and works well, therefore not modified during experiments. It is worth noting that there may exist better parameter configuration for the involved tasks, as we do not conduct exhaustive search on all combinations of parameter settings which is also intractable. As an unsupervised learning task, parameters are tuned on the training dataset.

## C Time and space complexity

The time and space complexities of Alg. 1 and Alg. 2 are analyzed in this section. Without loss of generality, we assume all graphs contain $n$ nodes for easy analysis. For GA-MGM (Alg. 1), considering the compact matrix form discussed in Sec. B.1, the update of $\mathbf{V}$ takes a chain of matrix multiplication where $\mathbf{A}, \mathbf{W} \in \mathbb{R}^{mn \times mn}, \mathbf{U} \in [0,1]^{mn \times d}$. This product should be efficiently computed using the chain order algorithm which selects the order with the lowest cost. Considering vanilla matrix multiplication ($\mathcal{O}(N^3)$ time complexity for two $N \times N$ matrices), the most efficient computing order for the update of $\mathbf{V}$ takes $\mathcal{O}(3m^2n^2d + 2mnd^2 + mnd)$, since $mn > d$ for MGM and MGMC problems, it is equivalent to $\mathcal{O}(m^2n^2d)$. The space complexity is $\mathcal{O}(m^2n^2)$ which is equal to the size of largest encountered matrix. For each $\mathbf{V}_i$, since $n \leq d$ and we add dummy variables, the Sinkhorn step takes $\mathcal{O}(d^2)$ each row/column normalization. By setting the largest number of iterations $K_{\text{SK}}$, the Sinkhorn step takes $\mathcal{O}(K_{\text{SK}}md^2)$ for $m$ graphs. The occupied space is $\mathcal{O}(md^2)$ during Sinkhorn iterations. When adding dummy nodes, the Hungarian projection takes $\mathcal{O}(md^3)$ time complexity and $\mathcal{O}(md^2)$ space complexity. Therefore, for a maximum of $K$ iterations in GA-MGM, the overall time complexity of GA-MGM (Alg. 1) is $\mathcal{O}(Km^2n^2d + K_1K_{\text{SK}}md^2 + K_2md^3)$, where $K_1, K_2$ are the number of times when Sinkhorn and Hungarian method is called, respectively, and $K_1 + K_2 = K$.

For our tested cases where $m$ is large and $d - n$ is relatively small, it holds $mn^2 \geq K_{\text{SK}}d$ and $mn^2 \geq d^2$, the time complexity of Alg. 1 can be simplified as $\mathcal{O}(Km^2n^2d)$, where $K$ is the max number of iterations. The space complexity of GA-MGM (Alg. 1) is $\mathcal{O}(m^2n^2)$. Similar analysis can be applied to GA-MGMC (Alg. 2), where the time complexity is $\mathcal{O}(K'Km^2n^2d)$ where $K'$ is the number of iterations in GA-MGMC, and the space complexity is $\mathcal{O}(m^2n^2)$. It is worth noting that the time complexity analyses are based on single-thread operations, and most operations in our proposed method are based on matrix computation, with which acceleration could be easily achieved with GPU. For example, our reported inference time in Tab. 2 grows little from the smaller-scaled problem to the larger-scaled problem. In comparison, other peer methods suffer from magnitudes of growth in inference time, when scale up to the larger problem.

# D  Experiment details

## D.1  Dataset details

**Willow Object Class**[1] [15] contains 304 images from 5 categories, collected from Caltech-256 [48] (208 faces, 50 ducks and 66 winebottles) and Pascal VOC 2007 [49] (40 cars and 40 motorbikes). Each image contains one object and is labeled with 10 common semantic keypoints with no outliers. It is worth noting that there are originally 209 face images, but the face image No. 0160 is labeled with only 8 keypoints (which is probably a mistake), and we exclude this image during evaluation.

**CUB2011 dataset**[2] [44] includes 11,788 bird images from 200 categories with closely 50%/50% split of training/testing images for each category, and each bird is labeled with a partial of 15 keypoints. The keypoints may be self-occluded which results in partial matching, the poses of birds may vary from flying, standing and swimming, and the images may contain different illumination and background situations. All these factors yield challenges to the matching task. In our experiment, only matching within the same category is considered.

## D.2  Computing infrastructure

Experiments are conducted on our Linux workstation with Xeon-3175X@3.10GHz, RTX8000, and 128GB memory.

## D.3  Data pre-processing

All images are resized to $256 \times 256$ and normalized, before passed to the VGG16 network. The raw RGB values are firstly divided by 256 (i.e. normalized to $[0, 1)$), and normalized by mean $[0.485, 0.456, 0.406]$ and STD $[0.229, 0.224, 0.225]$ which are collected from ImageNet statistics.

## D.4  Detailed results of Tab. 2

Detailed result of MGMC test on Willow Object Class dataset is listed in Tab. 5, 6, where both mean and STD statistics are shown. Tab. 5 contains the result with 8 Cars, 8 Ducks and 8 Motorbikes (in line with the left part of Tab. 5), and Tab. 6 contains the result with 40 Cars, 50 Ducks and 40 Motorbikes (in line with the right part of Tab. 6). Compared to peer methods, our methods are with higher STD, but still outperforms other competing methods. Our methods runs significantly faster among peer methods especially on the larger-sized problem, which is desired for real-world applications.

## D.5  Detailed results of Fig. 2

In the main paper, we only report the f1-score of MGM problem on CUB2011 dataset due to limited space. As shown in Fig. 3, here we provide more detailed result on the MGM problem of CUB2011 dataset, including matching precision, recall, and f1-score. Our unsupervised learning GANN-MGM surpasses GMN [16], and performs comparatively against novel supervised deep graph matching

Table 5: MGMC on Willow dataset with 8 Cars, 8 Ducks, 8 Motorbikes (mean and STD by 50 tests).

| method | learning | CP | RI | CA | MA | time (s) |
|---|---|---|---|---|---|---|
| RRWM [2] | free | 0.420±0.036 | 0.546±0.013 | 0.347±0.019 | 0.748±0.069 | **0.3** |
| CAO-C [10] | free | 0.435±0.044 | 0.549±0.017 | 0.352±0.024 | 0.875±0.059 | 2.7 |
| CAO-PC [10] | free | 0.445±0.043 | 0.552±0.015 | 0.357±0.021 | 0.867±0.065 | 1.4 |
| DPMC [19] | free | 0.435±0.044 | 0.549±0.016 | 0.352±0.023 | 0.886±0.082 | 1.0 |
| GA-MGMC (ours) | free | 0.921±0.058 | 0.905±0.063 | 0.893±0.063 | 0.653±0.146 | 10.6 |
| GANN-MGMC (ours) | unsup. | **0.976±0.037** | **0.970±0.045** | **0.963±0.051** | **0.896±0.072** | 5.2 |

Table 6: MGMC on Willow w/ 40 Cars, 50 Ducks, 40 Motorbikes (mean and STD by 50 tests).

| method | learning | CP | RI | CA | MA | time (s) |
|---|---|---|---|---|---|---|
| RRWM [2] | free | 0.592±0.000 | 0.681±0.002 | 0.556±0.003 | 0.751±0.000 | **8.8** |
| CAO-C [10] | free | 0.623±0.010 | 0.686±0.002 | 0.561±0.003 | 0.906±0.000 | 1051.5 |
| CAO-PC [10] | free | 0.665±0.006 | 0.695±0.003 | 0.572±0.005 | 0.886±0.000 | 184.0 |
| DPMC [19] | free | 0.585±0.001 | 0.653±0.000 | 0.511±0.000 | **0.941±0.001** | 97.5 |
| GA-MGMC (ours) | free | 0.890±0.060 | 0.871±0.061 | 0.850±0.061 | 0.669±0.122 | 11.7 |
| GANN-MGMC (ours) | unsup. | **0.974±0.034** | **0.968±0.035** | **0.956±0.039** | 0.906±0.047 | 9.2 |

Figure 3: Mean and STD of precision, recall and f1-score of MGM and supervised two-graph matching on CUB2011 dataset. MGM statics are computed from all graph pairs each category, and two-graph matching statics are computed from 1000 random graph pairs.

Figure 4: Study on the effectiveness of gradually decreasing the clustering weight $\beta$ from 1 to 0 for the MGMC problem. In this experiment, the 3-cluster MGMC problem on Willow dataset (40 Cars, 50 Ducks, 40 Motorbikes, in line with the right half of Tab. 2) is solved, and clustering and matching metrics are plotted with respect to the number of iterations in GA-MGMC. In this run, $\beta$ drops from 1 to 0.9 at iteration 3, and drops from 0.9 to 0 at iteration 6. If both matching and clustering do not change, it means that both of them have converged and $\beta$ is declined at next iteration. When $\beta$ declines, the matching accuracy improves because more confidence is counted on the clustering result, and the clustering accuracy also improves when more accurate matching is obtained.

methods PCA-GM [17] and CIE [34] on all matching metrics. The reason why our unsupervised learning GANN-MGM performs better on testing data than training data is probably because the testing set is slightly easier than the training set, as our learning-free version GA-MGM also performs better on testing data.

Figure 5: Ablation study on different initialization techniques with learning-free GA-MGMC on the 130 graphs-MGMC problem of Willow dataset (in line with right half of Tab. 2). Random initialization with different denominators are compared, together with GA-MGMC initialized by spectral matching [11].

Figure 6: Ablation study by different $\gamma$ values on the 130 graphs MGMC task on Willow dataset (in line with the right half of Tab. 2). The left image includes all clustering, matching, and inference time statistics, and we zoom in to focus on the matching accuracy (MA) in the right image.

## E Further study

### E.1 Validation of graduated assignment on MGMC problem

We validate that our GA-MGMC method works by gradual improvement on both matching and clustering, finally achieving satisfying matching and clustering results. In Fig. 4, we plot the clustering and matching metrics with respect to the iteration index, when a learning-free GA-MGMC model deals with the 130-graph MGMC problem on Willow Object Class dataset (in line with right half of Tab. 2). As shown in Fig. 4, the algorithm starts with no clustering information, resulting in a low-quality matching at the first iteration, with which a moderately good clustering is achieved. With decreased clustering weight $\beta$, more accurate clustering again improves matching accuracy and vice versa, finally reaching a satisfying matching and clustering result. Therefore, our GA-MGMC improves matching and clustering simultaneously with graduated assignment.

### E.2 Ablation study

Ablation studies are conducted on the MGMC problem on Willow dataset with 40 Cars, 50 Ducks, 40 Motorbikes, which is in line with the right half of Tab. 2.

For the random initialization of $\{\mathbf{U}_i\}$, we also experiment with other initialization techniques, e.g. initialize by spectral multi-matching [11]. As shown in Fig. 5 where different denominator configurations are compared with spectral matching, different initialization techniques perform

Table 7: Matching accuracy on Willow dataset (50 tests), where the backbone net of GA-MGM-PIA is built with PIA-GM [17].

| method | learning | car | duck | face | mbike | wbottle |
|---|---|---|---|---|---|---|
| GA-MGM (ours) | free | 0.746±0.153 | 0.900±0.106 | 0.997±0.021 | 0.892±0.139 | 0.937±0.072 |
| GA-MGM-PIA [17] | free | 0.380±0.103 | 0.434±0.153 | 0.484±0.102 | 0.450±0.161 | 0.421±0.064 |
| GANN-MGM (ours) | unsup. | 0.964±0.058 | 0.949±0.057 | 1.000±0.000 | 1.000±0.000 | 0.978±0.035 |
| GANN-MGM-PIA [17] | unsup. | 0.394±0.009 | 0.493±0.026 | 0.501±0.009 | 0.426±0.023 | 0.478±0.006 |

Table 8: MGMC on Willow w/ 40 Cars, 50 Ducks, 40 Motorbikes (mean and STD by 50 tests).

| method | learning | CP | RI | CA | MA | time (s) |
|---|---|---|---|---|---|---|
| GA-MGMC (ours) | free | 0.890±0.060 | 0.871±0.061 | 0.850±0.061 | 0.669±0.122 | 11.7 |
| GA-MGMC-PIA [17] | free | 0.607±0.102 | 0.645±0.068 | 0.514±0.102 | 0.261±0.060 | 17.9 |
| GANN-MGMC (ours) | unsup. | 0.974±0.034 | 0.968±0.035 | 0.956±0.039 | 0.906±0.047 | 9.2 |
| GANN-MGMC-PIA [17] | unsup. | 0.567±0.061 | 0.633±0.029 | 0.451±0.044 | 0.255±0.023 | 19.2 |

comparatively in all metrics. Therefore, our graduated assignment method is not sensitive to different initialization techniques, and we adopt random initialization for its cost-efficiency.

We also test different configurations of $\gamma$ from 0.5 to 0.95 at 0.05 interval. As shown in Fig. 6, the inference time decreases with respect to growing $\gamma$, and the matching accuracy (MA) peaks at $\gamma = 0.8$. For $\gamma < 0.6$, the graduated assignment method becomes hard to converge, resulting in relatively lower accuracy and more inference time. The clustering-related metrics do not change significantly with $\gamma$. Our selected $\gamma = 0.8$ achieves best average MA and moderately good clustering results with satisfying inference time.

### E.3 Feature extraction with GNN

Many recent efforts in learning deep graph matching networks involve learning graph structures with graph neural networks. Here we experiment our method with graph convolutional network, which can be viewed equivalent to replacing the VGG16 CNN in our pipeline by PIA-GM [17] (which contains a VGG16 and 3-layer GCN). The more powerful PCA-GM is not considered in our experiment because it seems nontrivial to define the cross-graph convolution operation in PCA-GM when jointly matching multiple graphs. In this experiment, the VGG16 net of PIA-GM is initialized with ImageNet weights and the GCN layers are randomly initialized for fair comparison with [17]. As shown in Tab. 7 and Tab. 8 where MGM and MGMC problems on Willow dataset are considered, respectively, the performance of methods involving PIA-GM are relatively poor compared to their GNN-free variants. A possible explanation is that initialization is important for unsupervised learning on GANN-MGM and GANN-MGMC, and random initialization may be inadequate for the GCN layers. Powerful unsupervised initialization techniques may be adopted for the GCN weights, and we think it may be beyond the scope of this paper and leave this for future work.

### E.4 Visualization of clustering result

A visualization of the prediction of GANN-MGMC is shown in Fig. 7, where our proposed approach correctly separates most images. The poses of birds do not vary much inside the ground truth cluster, as most Honored Grebes are swimming, most Baird Sparrows are standing and most Caspian Terns are flying. However, there also exist a flying grebe (3rd image on first row) and a standing tern (7th image on last row), and it is quite interesting that these two images are the only two misclassified images in this run, suggesting that our GANN-MGMC works well when distinguishing different poses, which is reasonable. Such phenomenon is probably caused by our manually defined graph-wise similarity metric for clustering (in Eq. 9), as the structural agreement term encourages to distinguish different graph structures (i.e. different poses), and the node-wise similarity term may care more about local image feature rather than the high-level image feature that is required for image classification tasks.

The visualization result in Fig. 7 suggests that with our matching-aware clustering measure (Eq. 9), our MGMC approach works by utilizing the structural information in graphs. It happens to work with this MGMC problem on CUB2011, where most birds from different categories can be distinguished by different graph structures. However, we also believe that achieving better clustering result among different bird categories with high-level image feature is beyond the scope of this paper, because it is

Figure 7: The visualization of clustering result on the MGMC problem of CUB2011 dataset. The images are separated in three rows according to their ground truth categories, and the color of the outer box (red/blue/yellow) shows the precited categories.

beyond the scope of graph information (i.e. node information and edge information) that is available for the most general MGMC task considered in this paper.

## Footnotes

[1]http://www.di.ens.fr/willow/research/graphlearning/WILLOW-ObjectClass_dataset.zip

[2]http://www.vision.caltech.edu.s3-us-west-2.amazonaws.com/visipedia-data/CUB-200-2011/CUB_200_2011.tgz