[Reviews · NeurIPS 2020]

Review 1

Summary and Contributions: The authors propose a graph matching method, which aligns the nodes of multiple graphs simultaneously. Moreover a clustering can be constructed performing the alignment only within the clusters. These hard optimization problems are solved by adapting the classical graduated assignment method for graph matching to the new settings.

Strengths: 1) The combination of multiple graph matching and clustering is really nice and has only recently been introduced. 2) The methods introduced are sound relying on well-established techniques and are well adapted to the new setting. 3) The empirical evaluation is extensive and clearly shows the merits of the approach.

Weaknesses: My main concern is that the graduated assignment method has been adapted to multi-graph matching before limiting the original contribution of the authors. Unfortunately, these publications (see below) are not discussed in the paper. Besides this (severe) issue, this is a good publication. The discussion of images in Sec. 2.1.1 comes unexpected. The application to image data should better be explained in the introduction.

Correctness: The proposed methods and claims are correct and the evaluation seems reasonable.

Clarity: Overall the paper is well-written and comprehensible. However, there are some language problems such as missing articles, which could be easily fixed. Minor remarks: - The sentence: "The converged S is the doubly-stochastic relaxation from the output of the Hungarian algorithm." does not make sense. S is a solution to the relaxed problem. - Eq (8): Since C is optimized, it should probably appear together with X_ij below \max.

Relation to Prior Work: Graph matching has been studied extensively in the literature. Unfortunately, the authors seem to have overlooked these publications extending the GA method to multi-graph matching: Albert Solé-Ribalta, Francesc Serratosa: Graduated Assignment Algorithm for Finding the Common Labelling of a Set of Graphs. SSPR/SPR 2010: 180-190 Albert Solé-Ribalta, Francesc Serratosa: Graduated Assignment Algorithm for Multiple Graph Matching Based on a Common Labeling. Int. J. Pattern Recognit. Artif. Intell. 27(1) (2013) The method proposed in the publications above appears to be closely related to Section 2.1 of the paper under review.

Reproducibility: Yes

Additional Feedback: === Update after the rebuttal === Thank you for pointing out the differences to the omitted references. However, in my opinion the original contribution of the MGM part is limited and the differences are not fully sufficient. It is not clear, if the main difference (highlighted in [red] in the feedback) really provides an advantage compared to adding dummy nodes as proposed and discussed in the omitted journal paper. The other differences are technical, but no major contributions in my opinion. The original contribution should be made clear on acceptance, which I cannot fully recommend for the reason stated above.


Review 2

Summary and Contributions: This work deals with a joint problem of graph matching and graph clustering. First, the authors proposed a new graph matching solver based on the first-order Taylor expansion of the Koopmans-Beckmann’s graph matching formulation. Then, the authors combined the multi-graph matching and clustering as an end-to-end model. Finally, they developed an unsupervised graph matching framework. Numerical experiments on Willows and CUB2011 datasets shows the proposed method reach a good performance in both non-learnable and learnable setting.

Strengths: The unsupervised graph matching model is particularly interesting and potentially has nice impact on many practical applications, where the manual labeling is expensive.

Weaknesses: The authors should test the proposed method on more challenging datasets, for example PASCAL VOC graph matching dataset, as the performance of recent proposed matching algorithms on Willows and CUB2011 is near saturated.

Correctness: The method and experiment are sound.

Clarity: The paper is well written, easy to read.

Relation to Prior Work: Yes.

Reproducibility: Yes

Additional Feedback: Update after reading the rebuttal letter: Thanks for the authors' feedback. I'd like to keep my score unchanged.


Review 3

Summary and Contributions: Authors propose a novel method for multi-graph matching that allows to backpropagate through the per-node feature extractors. The main novelty lies in utilizing the differentiable Sinkhorn algorithm for the bipartite matching part of the problem, and solving the multi-graph problem via Taylor expansion of the Koopmans-Beckmann’s QAP objective function. Results indicate SOTA performance on 2 datasets.

Strengths: + A good theoretical contribution that meaningfully combines several graph matching algorithms while maintaining backpropagability. + SOTA performance.

Weaknesses: - Similar to alternatives, it seems the method is quite memory heavy. - It is not obvious whether both Alg. 1 and Alg. 2 are guaranteed to converge.

Correctness: The paper seems theoretically correct.

Clarity: The paper is reasonably easy to follow.

Relation to Prior Work: The paper revises the prior work well. It builds on top of previous findings in a meaningful way.

Reproducibility: Yes

Additional Feedback: I like the chosen methodology and the problem in general. As the main disadvantage I view large memory consumption due to the need to keep all matching matrices in memory, which is the common issue with all alternatives. Also, it would be great to state whether both Alg. 1 and Alg. 2 are guaranteed to converge. Questions: 1) I wonder whether it is meaningful to setup A_i (the graph connectivities) using distances of 2D keypoints. Intuitively, these distances change as the object changes pose in front of the camera and such "distance matrix" thus does not contain information invariant to the pose of the object. 2) It would be great to conduct some experiments where one does not rely on matching GT annotated keypoints. MatchALS did such experiment on a dataset of cars. 3) What is the theoretical limit for the size of a dataset that could be matched using the proposed method? I presume that one can only match 100s of images with 10-100 keypoints in them. Can authors suggest a way to decrease the memory consumption? E.g. can the matching matrix be subsampled somehow and thus only a small random subpart of it will be considered during every iteration?


Review 4

Summary and Contributions: As a commonly used graph data processing technology, graph matching has application scenarios and research value in the field of machine learning. In this paper, the authors handle the multi-graph matching problem via a graduated assignment procedure. Furthermore, the authors devise the first unsupervised deep graph matching networks.

Strengths: 1. The authors propose a novel multi-graph matching solver by iteratively solving the first-order Taylor expansion of MGM Koopmans-Beckmann’s QAP. 2. The authors devise the first unsupervised deep graph matching network whose performance can be on par with supervised model for two graph matching. 3. Sufficient experimental results show the proposed approach effective.

Weaknesses: Even though I am not an expert in this field,I feel that the novelty of proposed method is somewhat Incremental. All the important formulas,such as Eq.(1) to Eq.(5), are referenced from the existing works,which makes the readers hard to know which the acutal contribution is.

Correctness: yes

Clarity: yes

Relation to Prior Work: I am not sure

Reproducibility: Yes

Additional Feedback: see weaknesses

[Author Response · NeurIPS 2020]

1　We would like to thank all reviewers and area chairs for help reviewing our submission. We are responding to the main
2　concerns raised by reviewers and we will update our final version accordingly.

3　**Novelty of Alg. 1 (R2).** Sorry for overlooking the mentioned graduated assignment multi-graph matching works [a, b]
4　which we find have also been cited in the references [9, 10]. We will add and discuss these two important papers in our
5　final version to better position our work. Below we elaborate their differences and compare the two algorithms directly.

| **Algorithm 1: GA-MGM (ours)** | **Algorithm 2: GA-CL [a, b]** i.e. the mentioned papers |
|---|---|
| Randomly initialize $\{\mathbf{U}_i\}$; projector ← Sinkhorn;<br>**while** *True* **do**<br>　**while** $\{\mathbf{U}_i\}$ *not converged* **do**<br>　　update $\mathbf{V}$ : $\mathbf{V} = \lambda \mathbf{A}(\mathbf{U}\mathbf{U}^\top \circ \mathbb{M})\mathbf{A}\mathbf{U} + (\mathbf{W} \circ \mathbb{M})\mathbf{U}$;<br>　　$\mathbf{U}_i \leftarrow$ projector$(\mathbf{V}_i, \tau)$;<br>　**if** projector == Sinkhorn & $\tau \geq \tau_{min}$ **then**<br>　　$\tau \leftarrow \tau \times \gamma$;<br>　**else if** projector == Sinkhorn & $\tau < \tau_{min}$ **then**<br>　　projector ← Hungarian;<br>　**else**<br>　　**break**; | Initialize a namely 'prototype' graph [a, b] for labeling<br>　the rest graphs: $\mathbf{U}_0 = \mathbf{I}$, and the rest graphs' matching<br>　to the 'prototype' $\{\mathbf{U}_i\}$ as random permutations;<br>**while** *True* **do**<br>　**while** $\{\mathbf{U}_i\}$ *not converged* **do**<br>　　update $\mathbf{V}$ : $\mathbf{V}[i, a, b] = \{\text{for ...}\} \times 8$;<br>　　$\mathbf{U}_i \leftarrow$ Sinkhorn$(\mathbf{V}_i, \tau)$;<br>　**if** $\tau \geq \tau_{min}$ **then**<br>　　$\tau \leftarrow \tau \times \gamma$;<br>　**else**<br>　　**break**; |
| **Output:** Joint matching matrices $\{\mathbf{U}_i\}$. | **Output:** Joint matching matrices $\{\mathbf{U}_i\}$. |

6　**1) [red]** First and foremost, the fundamental difference is that [a, b] require a 'prototype' graph as the anchor for labeling
7　the nodes of the rest graphs (in their paper they use the first graph as the 'prototype'). This means they assume the
8　bijection between each pair of graphs which is hard to satisfy in practice. In contrast, our method is fully decentralized
9　with no such constraint, and it can therefore handle the more practical and more challenging partial matching case, i.e.
10　only part of the nodes in each graph can find their correspondence from the other graphs. In fact, our method has one
11　more free variable $\mathbf{U}_0$ which is fixed to an identity matrix in [a,b]. Our method shows state-of-the-art performance on
12　CUB2011 dataset which involves partial matching, yet [a, b] are incapable for such challenging tasks by their design. **2)**
13　**[yellow]** Our method is clustering-aware by introducing the additional flag variable to account for clustering (also see
14　L6 in Alg. 1 in main paper) but [a, b] are not. **3) [blue]** Our method involves a more concise and compact formulation
15　based on matrix form, which leads to a GPU friendly implementation. This is important to large-scale QAP problem
16　especially for multiple graphs. This reformulation is nontrivial as shown in our supplemental material B.1 for the
17　detailed derivatives. **4) [green]** A few technical details are rather different, e.g. we adopt Hungarian method for early
18　discritization but the related papers contiuously use soft-assign which can be quite time consuming.

19　**About Pascal VOC (R3).** We do not compare on Pascal VOC because existing two-graph matching works are built
20　and tested under the setting that only the common keyponits in two graphs are filtered for matching. However, such a
21　setting does not apply to multi-graph matching, therefore our method cannot be fairly compared with existing works.

22　**About memory consumption (R4).** The memory consumption is still acceptable with our testbed (max 109 graphs, 10
23　nodes), however this issue may become severe if the problem size continues to scale up. We are aware of a recent work
24　[25] passing multi-graph matching information via a spanning tree, which may mitigate the memory issue suggested by
25　R4. And we think this issue may be beyond the scope of this paper and we would like to leave it as future work.

26　**About convergence of Alg. 1 and Alg. 2 (R4).** The convergence property of graduated assignment is discussed in
27　details by [c]. We cannot include detailed proof on convergence due to limited space, and only major steps are discussed.
28　We define $\mathbf{U}^k$, $\mathbf{U}^{(k+1)}$ as the (relaxed) joint matching matrix at k-th and (k+1)-th iteration respectively. By firstly fixing
29　some $\mathbf{U}^k$, the objective function in Alg. 1 can be proved to grow like an Expectation-Maximization (EM) algorithm:

$$\lambda \operatorname{tr}(\mathbf{U}^k \mathbf{U}^{k\top} \mathbf{A} \mathbf{U}^k \mathbf{U}^{k\top} \mathbf{A}) + \operatorname{tr}(\mathbf{U}^k \mathbf{U}^{k\top} \mathbf{W}) \leq \lambda \operatorname{tr}(\mathbf{U}^{(k+1)} \mathbf{U}^{k\top} \mathbf{A} \mathbf{U}^k \mathbf{U}^{k\top} \mathbf{A}) + \operatorname{tr}(\mathbf{U}^{(k+1)} \mathbf{U}^{k\top} \mathbf{W})$$
$$\leq \lambda \operatorname{tr}(\mathbf{U}^{(k+1)} \mathbf{U}^{(k+1)\top} \mathbf{A} \mathbf{U}^{(k+1)} \mathbf{U}^{(k+1)\top} \mathbf{A}) + \operatorname{tr}(\mathbf{U}^{(k+1)} \mathbf{U}^{(k+1)\top} \mathbf{W})$$

30　Similar conclusion holds for the multi-graph matching and clustering algorithm in Alg. 2.

31　**Questions by R4.** The set up of $\mathbf{A}_i$ follows [32]. We will include matching w/o GT in the final version if space permits.

32　**About our contributions (R5).** Eq. 1-5 are actually common formulations in this field and we list these formulations
33　to provide our readers a general picture if they are not familiar with graph matching. Our contributions also include a
34　graduated assignment algorithm for the more challenging multi-graph matching and clustering task (Sec. 2.2).

35　**[a]** A. Solé-Ribalta et al. Graduated assignment algorithm for finding the common labelling of a set of graphs, in *SSSPR* (2010).
36　**[b]** A. Solé-Ribalta et al. Graduated assignment algorithm for multiple graph matching based on a common labeling, *IJPRAI* (2013).
37　**[c]** A. Rangarajan et al. Convergence Properties of the Softassign Quadratic Assignment Algorithm, *Neural Computation* (1999).


[Meta-Review · NeurIPS 2020]

This paper presents a novel method that combines multi-graph matching with clustering. The method introduces several interesting components to achieve this goal, namely back-propagating through the Sinkhorn algorithm and solving the multi-graph problem with a first-order Taylor expansion of Koopmans-Beckmann’s QAP. As noted by reviewers, the paper needs some rewriting to better discuss previous work and clarify some of the claims, however, the contributions of this work are sufficiently important to recommend acceptance.